# The Effects of the Steroids 5-Androstenediol and Dehydroepiandrosterone and Their Synthetic Derivatives on the Viability of K562, HeLa, and Wi-38 Cells and the Luminol-Stimulated Chemiluminescence of Peripheral Blood Mononuclear Cells from Healthy Volunteers

**DOI:** 10.3390/biom14030373

**Published:** 2024-03-19

**Authors:** Mikhail N. Sokolov, Vladimir V. Rozhkov, Maria E. Uspenskaya, Darya N. Ulchenko, Vladimir I. Shmygarev, Vladimir M. Trukhan, Andrei V. Churakov, Nikolay L. Shimanovsky, Tatiana A. Fedotcheva

**Affiliations:** 1Institute for Translation Medicine and Biotechnology, Sechenov First Moscow State Medical University, 2 Bolshaya Pirogovskaya St., Bld. 4, Moscow 119435, Russia; sokolov_m_n@staff.sechenov.ru (M.N.S.); vladimir_rozhkov@mail.ru (V.V.R.); cheminfoinbox@gmail.com (V.I.S.); vladimir.trukhan@gmail.com (V.M.T.); 2Laboratory of Molecular Pharmacology, Pirogov Russian National Research Medical University, 1 Ostrovityanova St., Moscow 117997, Russia; m.uspenskaja@mail.ru (M.E.U.); motci@list.ru (D.N.U.);; 3Kurnakov Institute of General and Inorganic Chemistry, the Russian Academy of Sciences, 31 Leninsky Av., Moscow 119991, Russia; churakov@igic.ras.ru

**Keywords:** dehydroepiandrosterone, androstenediol, DHEA, 5-AED, K562, HeLa, Wi-38, PBMC, luminol-stimulated chemiluminescence

## Abstract

In order to evaluate the role of substituents at 3-C and 17-C in the cytotoxic and cytoprotective actions of DHEA and 5-AED molecules, their derivatives were synthesized by esterification using the corresponding acid anhydrides or acid chlorides. As a result, seven compounds were obtained: four DHEA derivatives (DHEA 3-propionate, DHEA 3-butanoate, DHEA 3-acetate, DHEA 3-methylsulfonate) and three 5-AED derivatives (5-AED 3-butanoate, 5-AED 3,17-dipropionate, 5-AED 3,17-dibutanoate). All of these compounds showed micromolar cytotoxic activity toward HeLa and K562 human cancer cells. The maximum cytostatic effect during long-term incubation for five days with HeLa and K562 cells was demonstrated by the propionic esters of the steroids: DHEA 3-propionate and 5-AED 3,17-dipropionate. These compounds stimulated the growth of normal Wi-38 cells by 30–50%, which indicates their cytoprotective properties toward noncancerous cells. The synthesized steroid derivatives exhibited antioxidant activity by reducing the production of reactive oxygen species (ROS) by peripheral blood mononuclear cells from healthy volunteers, as demonstrated in a luminol-stimulated chemiluminescence assay. The highest antioxidant effects were shown for the propionate ester of the steroid DHEA. DHEA 3-propionate inhibited luminol-stimulated chemiluminescence by 73% compared to the control, DHEA, which inhibited it only by 15%. These data show the promise of propionic substituents at 3-C and 17-C in steroid molecules for the creation of immunostimulatory and cytoprotective substances with antioxidant properties.

## 1. Introduction

Steroids are important physiological and pharmacological regulators of cell growth and survival: synthetic steroids are used for the treatment of sex-hormone-dependent cancer [1]. Steroid molecules are preferred for their rigidness and high ability to penetrate biological membranes [2]. In recent years, different DHEA derivatives have been synthesized as potent antiproliferative agents since this natural steroid scaffold can contribute to the potential cytotoxic activity [1,3,4]. Strong cytotoxic, apoptotic, and anti-inflammatory effects with wide therapeutic windows have been demonstrated for the A-homolactam derivative of D-homolactone androstane, synthesized from DHEA [5].

DHEA is a precursor of endogenous steroid hormones and plays a key role as a metabolic intermediate in the biosynthesis of androgen and estrogen sex hormones [6]. This steroid and its sulfate (DHEA-S) are among the most abundant circulating steroids in humans. Of all compounds, the DHEA-S concentration in human blood can reach up to 10 µM [7,8].

The biological role of DHEA remains obscure. At present, it is known with certainty that DHEA is responsible for the androgenic effects of adrenarche [9,10,11], is a weak estrogen [6,12,13], and acts as a neurosteroid and neurotrophin by producing important effects on the central nervous system [14,15,16]. The steroid is a potent uncompetitive inhibitor (with respect to NADP+ and glucose-6-phosphate) of mammalian G6PD, thereby reducing the availability of NADPH and the generation of ROS by NADPH-dependent enzymes and, consequently, decreasing NOX-dependent ROS production in various cell types [17,18]. DHEA also inhibits the activity of NADPH-dependent cytochrome P450, which transforms biologically inactive carcinogens (for example, 7,12-dimethylbenzanthracene) into their carcinogenic forms [19,20]. Therefore, DHEA suppresses the tumor initiation process as well. The dietary administration of DHEA reduces the development of liver preneoplastic foci [21].

DHEA has a strong antiglucocorticoid effect, but its mechanism is unclear. It is only known that DHEA does not bind to the glucocorticoid receptor [22]. According to A. Schwartz, the mechanism of the cytotoxic action of DHEA and its derivatives on cancerous cells is due to three main aspects [18], described below.

DHEA is a potent uncompetitive inhibitor of mammalian G6PD and inhibits NOX-dependent ROS production in various cell types. The accumulation of both DHEA and G6P in the cell drives the uncompetitive inhibition of G6PD to become irreversible, leading to reactive oxygen species (ROS)-mediated cell death [23,24].

DHEA is a potent ERβ agonist, and ERβ activation has been shown to antagonize the proliferative effects of ERα activation [25,26].

PPARα and PPARγ are activated by DHEA. The effect of anticancer therapy based on drug-targeted PPARs differs from, or even opposes, that based on three peroxisome proliferator-activated receptor homotypes and varies for different cancer types. Mostly, PPARα and PPARγ activation by different drugs leads to the death of tumor cells [22,27].

The DHEA metabolite 5-androstenediol (5-AED) is an intermediate in the biosynthesis of testosterone from DHEA and acts as a weak androgen and estrogen steroid hormone. 5-AED stimulates the immune response [28]. Taking this fact into account, the product has been investigated as a candidate for use as a radiation countermeasure. It elevates the level of circulating granulocytes and platelets in animals and humans and enhances survival in mice and non-human primates with acute radiation syndrome. 5-AED promotes the survival of irradiated human hematopoietic progenitors and causes elevations in the levels of circulating G-CSF and interleukin-6 (IL-6) [29,30,31].

There is very little information about the action of DHEA and 5-AED and their derivatives on the viability of different cell types.

In the present work, the ability of DHEA, 5-AED, and a series of their derivatives to act as cytotoxic and antioxidant agents, as well as the role of etherification in these effects, has been studied.

## 2. Materials and Methods

### 2.1. General Synthesis

DHEA (**1**), 99% pure, was purchased from Xi’an Sonwu Biotech Co., Ltd., Xi’an, China. All other reagents (97% purity) were purchased from BLDpharm (Shanghai, China). All solvents were HPLC grade and were obtained from Carlo Erba reagents (Val de Reuil Cedex, France) and Panreac Quimica SLU (Barcelona, Spain). The solvents and reagents were used without further purification. Flash chromatography was performed with Merck silica gel (40–60 μm). Analytical thin-layer chromatography (TLC) was carried out on 0.25 mm commercial silica gel plates (Merk, silica gel 60 F_254_). TLC plates were visualized by UV fluorescence at 254 nm or with a 10% water solution of potassium permanganate.

^1^H and ^13^C NMR spectra were recorded on Bruker Fourier 300 NMR spectrometers for solutions in CDCl_3_, with tetramethylsilane (TMS) as an internal standard. Chemical shifts and coupling constants are given in ppm and Hz, respectively. Optical rotations were determined at 589 nm (sodium D line) by using a Jasco DIP-360 polarimeter, and the melting points were measured on a Büchi B-540 apparatus (“Buchi Labortechnik AG”, Flawil, Switzerland). The melting points and optical rotations of the known compounds are identical to those of the previously described steroids. HRMS was carried out on a Xevo G2-XS QTof mass spectrometer (Wilmslow, UK).

Synthesis of 5-androstenediol (3S,10R,13S,17S)-10,13-dimethyl-2,3,4,7,8,9,10,11,12,13,14,15,16,17-tetradecahydro-1H-cyclopenta-[a]phenanthrene-3,17-diol (2). To a solution of 1 (14.00 g, 0.049 mol) in THF (150 mL), NaBH_4_ (2.38 g, 0.062 mol) was added at room temperature with stirring. After that, 10 mL of water was added to the reaction mixture. The reaction mixture was stirred for 1 h, then 500 mL of water was added, and stirring was continued for an additional 2 h until a white solid formed. The precipitate was collected by filtration, washed with water, and dried in vacuo to furnish product 2. 

Yield: 14.00 g (99%). [α]_D_^25^ = −53.4° (c = 0.5; 2-propanol), mp = 179–180 °C (methanol) (lit. mp = 183 °C [32]).

^1^H NMR (300 MHz, MeOD) δ: 5.36 (d, *J* = 4.7 Hz, 1H), 3.59 (t, 1H), 3.46–3.37 (m, 1H), 2.28–2.21 (m, 2H), 2.08–1.74 (m, 5H), 1.69–1.17 (m, 8H), 1.16–0.87 (m + s, 7H), 0.77 (s, 3H).

^13^C NMR (75 MHz, MeOD) δ: 142.34, 122.24, 82.47, 72.41, 52.72, 51.92, 43.85, 43.04, 38.59, 37.90, 37.77, 33.33, 32.64, 32.30, 30.66, 24.37, 21.83, 19.91, 11.52.

Preparation of **3a** and **3c.** To a solution of DHEA (**1**) (2 g, 0.007 mol) in pyridine (6 mL) and toluene (10 mL), acetic anhydride (1.06 g, 0.011 mol, 0.98 mL) (for **3a**) or butyric anhydride (1.64 g, 0.011 mol, 1.70 mL) (for **3c**) was added at room temperature. The mixture was stirred at the same temperature overnight and poured into an ice-cooled 10% solution of HCl (100 mL). The organic layer was separated, washed with water and brine, and dried over Na_2_SO_4_. The solvent was evaporated, and the residue was recrystallized from hexane to furnish products **3a** and **3c** as white solids. The melting points and optical rotations of the synthesized compounds are identical to those of the previously described steroids. 

Dehydroepiandrosterone-3-acetate (3S,10R,13S)-10,13-Dimethyl-17-oxo-2,3,4,7,8,9,10,11,12,13,14, 15,16,17-tetradecahydro-1H-cyclopenta[a]phenanthren-3-yl acetate (**3a**) [26]. Yield: 1.95 g (85%). [α]_D_^25^ = −24.1 (c = 0.5; 2-propanol), mp = 170 (dec.) °C (hexane) (lit. [α]_D_^25^ = −23 (c = 0.5) mp = 172 °C [33]).

^1^H NMR (300 MHz, CDCl_3_) δ: 5.37 (d, *J* = 5.0 Hz, 1H), 4.57 (m, 1H), 2.47–2.18 (m, 3H), 2.15–1.74 (m + s, 9H), 1.71–1.35 (m, 6H), 1.33–0.93 (m + s, 7H), 0.85 (s, 3H).

^13^C NMR (75 MHz, CDCl_3_) δ: 220.89, 170.49, 140.01, 121.92, 73.76, 51.78, 50.24, 47.56, 38.15, 37.01, 36.80, 35.88, 31.55, 31.50, 30.84, 27.77, 21.94, 21.44, 20.39, 19.39, 13.61.

Dehydroepiandrosterone-3-butyrate (3S,10R,13S)-10,13-Dimethyl-17-oxo-2,3,4,7,8,9,10,11,12,13,14, 15,16,17-tetradecahydro-1H-cyclopenta[a]phenanthren-3-yl butyrate (**3c**) [27]. Yield: 2.00 g (80%). [α]_D_^25^ = −1.1 (c = 0.09; CH_2_Cl_2_), mp = 162–164 °C (hexane) (lit. [α]_D_ = 0 (c = 0.11), mp = 164.5–165 °C [34]).

^1^H NMR (300 MHz, CDCl_3_) δ: 5.39 (d, *J* = 4.9 Hz, 1H), 4.60 (m, 1H), 2.59–1.77 (m + t, *J_t_* = 7.3 Hz, 11H), 1.74–1.38 (m, 8H), 1.35–1.38 (m, 6H), 1.32–0.88 (m + s + t, *J_t_* = 7.3 Hz, 10H), 0.87 (s, 3H).

^13^C NMR (75 MHz, CDCl_3_) δ: 221.04, 173.20, 140.14, 121.91, 73.53, 51.84, 50.29, 47.62, 38.25, 37.08, 36.87, 36.67, 35.94, 31.61, 31.55, 30.90, 27.87, 21.89, 20.45, 19.46, 18.64, 13.74, 13.66.

Preparation of **3b** and **3d.** To a solution of DHEA (**1**) (2 g, 0.007 mol) in CH_2_Cl_2_ (20 mL) and Et_3_N (2 g, 0.02 mol, 1.93 mL), propionyl chloride (0.77 g, 0.008 mol, 0.73 mL) (for **3b**) or methanesulfonyl chloride (0.95 g, 0.008 mol, 0.64 mL) (for **3d**) was added at −50 °C. The mixture was allowed to warm up to room temperature and stirred overnight. The organic layer was washed with 10 mL of a 1% solution of HCl and brine and dried over Na_2_SO_4_. The solvent was evaporated, and the residue was recrystallized from hexane to furnish products **3b** and **3d** as white solids.

Dehydroepiandrosterone-3-propionate (3S,10R,13S)-10,13-Dimethyl-17-oxo-2,3,4,7,8,9,10,11,12,13,14, 15,16,17-tetradecahydro-1H-cyclopenta[a]phenanthren-3-yl propionate (**3b**) [27]. Yield: 1.95 g (82%). [α]_D_^25^ = −4.92 (c = 0.2; CH_2_Cl_2_), mp = 163–164 °C (hexane) (lit. [α]_D_ = −4.26 (c = 0.09) mp = 164.5–165 °C [34]).

^1^H NMR (300 MHz, CDCl_3_) δ: 5.39 (d, *J* = 4.9 Hz, 1H), 4.60 (m, 1H), 2.51–2.22 (m + q, *J_q_* = 7.7 Hz, 5H), 2.17–1.77 (m, 6H), 1.73–1.38 (m, 6H), 1.36–0.95 (m + t + s, *J_t_* = 7.7 Hz, 10H), 0.87 (s, 3H).

^13^C NMR (75 MHz, CDCl_3_) δ: 220.91, 173.88, 140.01, 121.79, 73.47, 51.71, 50.16, 47.50, 38.11, 36.94, 36.73, 35.81, 31.48, 31.42, 30.77, 27.89, 27.72, 21.86, 20.32, 19.33, 13.53, 9.15.

Dehydroepiandrosterone-3-methanesulfonate (3S,10R,13S)-10,13-Dimethyl-17-oxo-2,3,4,7,8,9,10,11,12,13,14, 15,16,17-tetradecahydro-1H-cyclopenta[a]phenanthren-3-yl methanesulfonate (**3d**)**.** Yield: 2.05 g (81%). [α]_D_^25^ = −6.87 (c = 0.1; CH_2_Cl_2_), mp = 162– 168 (dec.) °C (hexane) (lit. mp = 165–170 (dec.) °C) [35].

^1^H NMR (300 MHz, CDCl_3_) δ: 5.44 (d, *J* = 5.1 Hz, 1H), 4.51 (m, 1H), 3.00 (s, 3H), 2.58–2.37 (m, 3H), 2.18–0.93 (m + s, 19H), 0.87 (s, 3H).

^13^C NMR (75 MHz, CDCl_3_) δ: 220.85, 139.08, 123.16, 87.73, 51.78, 50.18, 47.60, 39.27, 39.91, 36.96, 36.65, 35.92, 31.53, 31.49, 30.88, 29.01, 21.97, 20.45, 19.34, 13.66.

Synthesis of 5-androstenediol-3,17-dipropionate (3S,10R,13S,17S)-10,13-dimethyl-2,3,4,7,8,9,10,11,12,13,14,15,16,17-tetradecahydro-1H-cyclopenta-[a]phenanthrene-3,17-diyl dipropionate (**4b**) [36]. To a solution of product **2** (2 g, 0.007 mol) in pyridine (27 mL), propionyl chloride (2.55 g, 0.028 mol, 2.39 mL) was added at −5 °C. The reaction mixture was warmed up to room temperature and stirred overnight. The mixture was portioned between toluene (20 mL) and water (25 mL), and the organic layer was separated, washed with brine, and dried over Na_2_SO_4_. The solvent was evaporated, and the residue was chromatographed on silica gel using hexane/ethyl acetate (10:1) as an eluent. Yield: 2.55 g (92%). [α]_D_^25^ = −11.8 (c = 0.5; CH_2_Cl_2_), mp = 127–129 °C (hexane)**.** Calculated monoisotopic mass for C_25_H_38_O_4_: 402.2770; found: m/z = 402.2765 [M + H]^+^.

^1^H NMR (300 MHz, CDCl_3_) δ: 5.35 (d, *J* = 4.9 Hz, 1H), 4.58 (m, 2H), 2.39–2.07 (m, 7H), 2.06–0.86 (m + s, 25H), 0.78 (s, 3H). 

^13^C NMR (75 MHz, CDCl_3_) δ: 174.61,173.97, 139.94, 122.26, 82.59, 73.11, 51.13, 50.08, 42.54, 38.23, 37.10, 36.87, 36.76, 31.80, 31.56, 28.00, 27.91, 27.85, 27.68, 23.70, 20.62, 19.44, 12.03, 9.38, 9.27.

Synthesis of 5-androstenediol-3,17-dibutyrate (3S,10R,13S,17S)-10,13-dimethyl-2,3,4,7,8,9,10,11,12,13,14,15,16,17-tetradecahydro-1H-cyclopenta-[a]phenanthrene-3,17-diyl dibutyrate (**4c**) [36]. To a solution of 5-AED (**2**) (2 g, 0.007 mol) in pyridine (6 mL) and toluene (10 mL), butyric anhydride (3.27 g, 0.021 mol, 3.38 mL) was added at room temperature. The mixture was stirred at the same temperature overnight and poured into an ice-cooled 10% solution of HCl (100 mL). The organic layer was separated and washed with water and brine and dried over Na_2_SO_4_. The solvent was evaporated, and the residue was recrystallized from hexane to furnish product **4c** as a white solid. Yield: 2.28 g (77%). [α]_D_^25^ = −15.6 (c = 0.1; CH_2_Cl_2_), mp = 130–132 °C (hexane) (lit. mp = 180 °C) [36].

^1^H NMR (300 MHz, CDCl_3_) δ: 5.36 (d, *J* = 4.8 Hz, 1H), 4.61 (m, 2H), 2.34–2.07 (m, 7H), 2.05–0.88 (m + s, 29H), 0.79 (s, 3H).

^13^C NMR (75 MHz, CDCl_3_) δ: 173.83, 173.20, 139.96, 122.27, 82.56, 73.66, 51.15, 50.11, 42.55, 38.27, 37.13, 36.89, 36.76, 36.69, 36.62, 31.83, 31.58, 27.90, 27.70, 23.72, 20.64, 19.46, 18.70, 18.65, 13.78, 13.74, 12.08.

Reduction of steroid **3c.** To a solution of **3c** (4.10 g, 0.011 mol) in THF (50 mL), sodium borohydride (0.56 g, 0.014 mol) was added at room temperature with stirring. Thereafter, water (5 mL) was added, and the reaction mixture was stirred for 1 h. Then, water (250 mL) and toluene (50 mL) were added, and the mixture was stirred for 15 min. The organic layer was separated, washed with brine, and dried over Na_2_SO_4_. The solvent was evaporated, and the residue was chromatographed on silica gel using hexane/ethyl acetate as an eluent. 

5-androstenediol-3-butyrate (3S,10R,13S,17S)-17-Hydroxy-10,13-dimethyl-2,3,4,7,8,9,10,11,12,13,14,15,16,17-tetradecahydro-1H- cyclopenta[a]phenanthren-3-yl butyrate (**5**) [30]. Yield: 3.64 g (88%). [α]_D_^25^ = −0.7 (c = 0.5; CHCl_3_), mp = 94–96 °C (hexane) (lit. [α]_D_^25^ = −0.6 (c = 0.5; CHCl_3_), mp = 90–92 °C) [37].

^1^H NMR (300 MHz, CDCl_3_) δ 5.36 (d, *J* = 5.0 Hz, 1H), 4.72–4.52 (m, 1H), 3.63 (t, *J* = 8.5 Hz, 1H), 2.38–2.17 (m + t, *J_t_* = 7.5 Hz, 4H), 2.14–0.82 (m + t, *J_t_* = 7.5 Hz, 26H), 0.75 (s, 3H).

^13^C NMR (75 MHz, CDCl_3_) δ: 173.26, 139.95, 122.36, 81.93, 73.70, 51.41, 50.30, 42.85, 38.28, 37.17, 36.80, 36.69 (2C), 32.05, 31.61, 30.60, 27.92, 23.55, 20.76, 19.48, 18.65, 13.74, 11.08.

#### Crystallographic Details

Experimental data sets were collected on a Bruker SMART APEX II diffractometer (for **2**) and a Bruker D8 Venture machine (for **3d**) using graphite monochromatized Mo-*K*α radiation (λ = 0.71073 Å). Absorption corrections based on the measurements of equivalent reflections were applied [38]. The structures were solved by direct methods and refined by full-matrix least-squares on *F*^2^ with anisotropic thermal parameters for all non-hydrogen atoms [39]. In structure **2**, all carbon H atoms were placed in calculated positions and refined using a riding model. Hydroxy atoms H1, H2, H11, H12, H21, and H22 in structure **2** were found from a difference Fourier map, and their positional parameters were freely refined. As for **3d**, all hydrogen atoms were found by the difference Fourier synthesis and refined with isotropic thermal parameters. The details of X-ray studies are listed in Appendix A. Single-crystal X-ray diffraction studies were performed at the Centre of Shared Equipment of IGIC RAS. The crystallographic data for **2** and **3d** were deposited in the Cambridge Structural Database under the numbers 2290975 and 2290974, respectively.

### 2.2. Biological Tests

#### 2.2.1. Cell Lines

The cytotoxic effects of the synthesized steroids on HeLa cervical cancer cells, lymphoblast cells K562, the normal Caucasian fibroblast-like fetal lung cell line Wi-38, and peripheral blood mononuclear cells (PBMCs) from healthy volunteers were assessed using the 3-(4,5-dimethylthiazol-2-yl)-2,5-diphenyltetrazolium bromide (MTT) test. The cells were incubated for 24 and 120 h (HeLa, Wi-38) with the synthesized compounds at a concentration of 10 µM. As a control, cells incubated in the presence of the solvent dimethyl sulfoxide (DMSO, PanEco, Moscow, Russia) at a concentration equivalent to that of the corresponding steroid were used.

HeLa, K562, and Wi-38 cell cultures were obtained from the unique scientific facility “Biocollection of FGBNU VILAR”. Cell cultivation was carried out under sterile conditions using a laminar box LB-V (Moscow, Russia). Cells were incubated at 37 °C in 5% CO_2_. The cells were grown using standard Dulbecco’s modified eagle medium (DMEM, Gibco, London, UK) supplemented with 10% heat-inactivated fetal calf serum Gibco, Auckland, New Zealand), L-glutamine at a concentration of 100 μg/mL, and the antibiotics gentamicin sulfate and streptomycin sulfate at a concentration of 40 μg/mL.

#### 2.2.2. MTT Assay

Cells were grown in 25 mL flasks; after the formation of the monolayer, trypsinization was performed, and 200 μL was added to the wells of a COSTAR flat-bottom plate (Corning, NY, USA). Solutions of the compounds were added to final concentrations of 10^−5^ M and incubated for 24 h. The DMSO concentration did not exceed 0.01%, and the control wells contained an equal volume of DMSO at each point. Thereafter, culture viability was assessed using a standard MTT assay [40]. After the completion of the incubation of cells with the compounds, the medium was taken from the wells of the plate, after which 200 µL of DMEM F12 1:1 medium and 10 µL of the stock MTT solution (10 mg/mL, Diam, Moscow, Russia) were added to the wells. Cells were incubated at 37 °C for 3 h in a humidified atmosphere of 5% CO_2_. After the completion of the incubation, the medium was removed from the wells, and 150 µL of DMSO was added to each well to dissolve the formed formazan crystals. The salt was dissolved for 15 min by shaking the plate at room temperature. Color development was recorded by determining the optical density at a wavelength of 530 nm on a plate photometer (UNIPLAN analyzer of enzyme immunoassay reactions AIFR-01, Ryazan, Russia). The ratio of the average optical density for a given concentration of a substance to the average optical density of the control was taken as the proportion of surviving cells.

#### 2.2.3. Participants

PBMCs were obtained from volunteers, postgraduate students of the Pirogov Russian National Research Medical University, who were healthy women in the age range of 20–22 (n = 6). Informed consent was obtained from all participants. No participants were on medications for hormonal replacement therapy, contraceptives, or non-steroidal anti-inflammatory drugs (NSAIDs). The women were non-smokers, with an average body mass index of 22.1 ± 3.1, without serious comorbidities. The study was conducted in accordance with the Declaration of Helsinki and approved by the Institutional Review Board of the Pirogov Russian National Research Medical University (No. 5/2023) on 10 May 2023. PBMCs were obtained by the combination of centrifugation and sedimentation at 1 g based on sedimentation in a single-stage Ficoll density gradient [41].

#### 2.2.4. LSCL Measurement

The luminol-stimulated chemiluminescence (LSCL) was measured using a Lum-100 chemiluminometer (DISoft, Moscow, Russia). The data obtained were evaluated using the PowerGraph 3.3 Professional software (DISoft, Moscow, Russia).

A control sample consisting of 100 µL of PBMCs was brought to 750 µL with Hank’s solution (pH = 7.45) and incubated at 37 °C for 45 min. Next, the sample was added to the cuvette of the chemiluminometer, and 150 µL of a luminol solution (Sigma-Aldrich, Saint Louis, MO, USA) at a concentration of 4.5 mM was added (the final concentration in the sample was 0.56 mM). Spontaneous kinetics were recorded for 3 min, after which 300 µL of an activator of the oxidative activity of cells (a luminescence stimulator, barium sulfate, VIPS-MED Firm, Moscow, Russia) was added at a concentration of 34.3 mM (final concentration in the sample was 8.6 mM), and the kinetics of LSCL were recorded for 10 min. The final sample volume was 1200 µL. The chemiluminescence was measured at 37 °C for 10 min. The slope of the chemiluminescence curve recorded for 5 min was used to characterize the rate of ROS production. Chemiluminescence was measured immediately after the addition of the drugs to the freshly obtained PBMCs.

#### 2.2.5. Statistical Analysis

Each MTT test experiment was repeated three times, with 3 repetitions (3 wells) within each experiment and 12 repetitions for the control wells (for the control in each experiment, the whole line of the 96-well plate was used to avoid large differences in the control points). The mean ± standard deviation value was calculated for each point.

The statistical significance of the LSCL data was determined using the Mann–Whitney U test (*p* < 0.05) between the control group (non-treated cells) and the treated group (drug-treated PBMCs). Each experiment was repeated three times. The mean ± standard deviation value was calculated for each point.

## 3. Results

### 3.1. Synthesis

All steroid compounds were obtained according to Figure 1.

Steroid **1** was reduced to diol **2** in 99% yield. The reaction proceeded stereospecifically, giving only one isomer, **2**. The process was carried out in aqueous tetrahydrofuran utilizing sodium borohydride as a reducing reagent. Monoester **5** was obtained by the same method. Compound **5** was already synthesized enzymatically using 2,2,2-trifluoroethyl butyrate to obtain this compound in 88% yield [30].

The slow evaporation of CD_3_OD from an NMR tube afforded diffraction-quality crystals. The S-configuration of position 17 was unambiguously confirmed by X-ray analysis. In both independent molecules of **2**, all bond lengths and angles possessed ordinary values for organic compounds. A single-crystal X-ray analysis revealed strong intermolecular hydrogen bonding between two 3-hydroxy groups in the structure of **2** (Figure 1). In the crystal, the molecular layers of **2** are additionally stabilized by hydrogen bonding between the 3-OH group and the molecules of the solvent CD_3_OD (Figure 2).

Esters **3** and **4** were synthesized with high yields utilizing either anhydrides or the chloroanhydrides of the corresponding acids. Pyridine (in the case of anhydrides) or triethylamine (in the case of chloroanhydrides) was used as a scavenger. The esterification products were purified by recrystallization from hexane (**3a**–**d**, **4c**) or by column chromatography (**4b**, **5**). To our surprise, the application of thiethylamine for the synthesis of ester **4b** gave a complex mixture of products. To obtain the title **4b**, we were forced to replace Et_3_N with pyridine. In this case, the reaction proceeded smoothly, giving **4b** in 92% yield.

The molecular structure and absolute configuration of the C-3β position of DHEA mesylate (**3d**) were also confirmed by the X-ray analysis (Figure 3).

### 3.2. Biological Tests

#### 3.2.1. The Effect of the Steroid Hormones 5-AED and DHEA and Their Derivatives on the Viability of K562, HeLa, and Wi-38 Cells

There are few publications regarding the effect of androstenes on the viability of various cell cultures. However, some evidence indicates that DHEA at a concentration of 100 µM inhibits the viability of A549, Hela, HepG-2, BEL7402, HCT116, MCF-7, and L02 cell cultures by 24.1, 5.7, 10.9, 46.5, 47.0, 17.1, and 44.6%, respectively. That is, the IC_50_ was not achieved in this study because a fixed concentration of DHEA was used [1]. In another study, the IC50 values were found to be 2.55 μM for T47D and 46.5 μM for Jurcat, and for the cultures MDA-MB-231, MCF-7, DU145, LNCaP, HCT116, HT29, and HL-60, the IC_50_ values were above 50 μM, with an incubation time of 96 h [42].

Based on the previously obtained data on IC_50_ in various cell cultures for steroids with different substituents at the C3 carbon atom of the cyclopentane perhydrophenanthrene ring, which ranged from 7 to 87 μM with 48 h incubation [40,43], we used a fixed concentration of steroids of 10 μM for the primary screening of cytotoxicity.

The cytotoxic effect of androstenes on tumor cells was compared with their effect on normal cells: Wi-38, human fetal fibroblasts, and PMBC of healthy volunteers. The data are shown in Table 1.

Androstenes are low-toxicity compounds; according to the Drugbank database, the acute oral toxicity of DHEA (LD_50_) is less than 10,000 mg/kg for *Rattus norvegicus*, and the lowest published dose for humans is 10 mg/kg [44]. In the United States, DHEA and DHEA-S have been believed to be beneficial for a wide variety of ailments, mainly related to aging. DHEA and DHEA-S are readily available in the United States, where they are marketed as over-the-counter dietary supplements. In November 2016, DHEA was approved (as Intrarosa) to treat women experiencing moderate to severe pains during sexual intercourse (dyspareunia), a symptom of vulvar and vaginal atrophy (VVA), due to menopause. The absence of a cytotoxic effect of androstenes on normal Wi-38 and PMBC cells (Table 1) confirms that the application of DHEA in these cases is justified.

The stimulating effect of glucocorticoids (e.g., hydrocortisone), another class of steroid hormones, on Wi-38 was demonstrated back in 1979, and the stimulation of proliferation was found to correlate with the number of glucocorticoid receptors [45]. The addition of 1.4 × 10^−7^ M cortisol to Wi-38 human fibroblasts led to an increase in proliferative activity in terms of the number of cells and the rate of incorporation of [^3^H]-thymidine into DNA. The precise mechanism of this stimulating effect has not been identified, but there are some studies showing that the extended lifespan of human Wi-38 cells that occurs when these cells are maintained in a culture medium supplemented with another glucocorticoid, dexamethasone, is accompanied by the suppression of p21(Waf1/Cip1/Sdi1) levels, which normally increase as these cells enter senescence, while p16(INK4a) levels are unaffected [46]. The anti-aging and regenerating effects of DHEA and AED are well known, but their mechanisms of action still remain unidentified. They have a regenerating anabolic effect on the muscle tissue and skin [47,48], neuroprotective and neurogenic influences [49,50], and immunostimulatory properties [51].

Previously, in agreement with the above-mentioned data about the low toxicity of androstenes to normal cells, our laboratory showed that 5-AED and DHEA at a concentration of 10 μM do not affect the viability of rat skin fibroblasts [52].

On the other hand, almost all androstenes were shown to inhibit the viability of cancerous K562 cells by 50%, whereas DHEA was less effective, inhibiting the cell viability only by 30% (Table 1). The cytotoxic effect of androstenes on lymphoid tissue cells has been revealed for the first time, which shows the prospects for the further study of these steroids as analogs of glucocorticoids in the therapy of lymphocytic leukemia.

Androstene derivatives and the endogen steroids DHEA and 5-AED themselves had a pronounced cytotoxic effect on HeLa cells (Table 1). The cytotoxic effect increased during the incubation time, suggesting a genomic receptor mechanism of action. Within 24 h, only two androstenes inhibited the viability of HeLa cells by 20%: DHEA 3-propionate (**3b**) and 5-AED 3,17-propionate (**4b**); but after more prolonged incubations, all steroids tested showed a cytotoxic effect (Table 1).

#### 3.2.2. The Effect of 5-AED and DHEA and Their Derivatives on the Luminal-Dependent Chemiluminescence of PBMCs from Healthy Volunteers

The multidirectional effects of various classes of steroid hormones on the immune system depend primarily on the presence of receptors on immune cells and their ability to bind to the hormones [53]. Here, we assessed the effects of 5-AED, DHEA, and their derivatives on the viability and oxidative capacity of macrophages from healthy volunteers to understand whether they have potential anti-inflammatory activity [54,55].

Androstenes were added to freshly obtained PBMCs from healthy volunteers, and the curves of chemiluminescence were recorded for 5 min. The slopes of the curves were normalized to the control sample, defined as 100% (Figure 4).

Luminol-dependent chemiluminescence is a sensitive method for the detection of antioxidant and anti-inflammatory activity [56]. The strongest inhibitory effect among all steroids tested was demonstrated by DHEA 3-propionate (**3b**).

## 4. Discussion

The aim of the current research was the synthesis of derivatives of 5-AED and DHEA to evaluate their possible roles as antitumor and anti-inflammatory agents. In addition, the role of esterification at 3-C and 17-C in steroid molecules in their cytotoxic effects has been scarcely investigated.

The analysis of publications from the PUBMED and GOOGLE SCHOLAR databases about the influence of DHEA on the viability of the HeLa, K562, and Wi-38 cell cultures revealed quite contradictory data, and information on the influence of 5-AED on the viability of cell cultures is nonexistent (Table 2).

The difference in the action of DHEA can be explained by the fact that steroids are metabolized in different ways in different cell types. So, HeLa cells show heterogeneity in the metabolism of the labeled hormone radiotestosterone by the alternative 17-oxosteroid and Δ4 pathways—a metabolic pathway that differs from that in prostate cancer cells [66]. Consequently, it is the metabolites of steroids that have the final effect, which proves the idea that the action of androstenes depends on the representation of cytochromes and steroid receptors (AR, ER, GR, and others) in the cell. On the other hand, DHEA strongly inhibits the proliferation of cervical cancer cells HeLa, but its effect is not mediated by androgen or estrogen receptor pathways, since the antiproliferative effect was not abrogated by the inhibitors of these receptors in these cells [62].

Table 2 demonstrates the inhibitory effect of DHEA on HeLa cells and the stimulating action on AR-negative and ER-positive T47D cells. At the same time, some DHEA derivatives had an IC50 similar to that of etoposide in the range of micromolar concentrations [67]. The 3-chloro-benzylidene derivative of DHEA, compound 1b, was the most potent synthesized derivative, especially against the KB and T47D cell lines (IC50 values were 0.6 and 1.7 μM, respectively), which is comparable to the action of etoposide (IC50 = 2.8 and 1.2 μM, respectively) [67]. In our study, DHEA inhibited the viability of HeLa cells by 20% after 24 h of incubation, and by the 5th day of incubation, the cytotoxic effect had disappeared. DHEA derivatives, on the contrary, after 5 days of incubation, had a more pronounced cytotoxic effect than DHEA, inhibiting the viability by more than 50%, on average by 50–80%.

The steroids studied had no influence on PBMC viability, which is in good accordance with previously published results [65]. Liu et al. [68] showed a cytoprotective effect of DHEA on endothelial cells, which was estrogen receptor-independent. They also showed that DHEA binds to specific receptors on the plasma membranes of endothelial cells, and that these receptors activate intracellular G proteins (specifically Gai2 and Gai3) and endothelial nitric oxide synthase. The absence of competition between DHEA and sulfated DHEA suggests that 3-position structures such as sulfated DHEA, androstenedione, 17a-hydroxypregnenalone, testosterone, and 17b-estradiol did not compete, and the A-ring may be an important component of the functional group for this receptor [68].

About 40 years ago, the influence of DHEA on the proliferation of Wi-38 cells was investigated. At that time, the IC50 parameter had not yet been determined; the method commonly used consisted of calculating the cells, and the incubation of cells took 16 days. It was shown that DHEA at a concentration of 137 μM decreased the number of Wi-38 cells by up to 65%, and at 18 μM, by up to 19%, while lower concentrations had no influence on cell growth. At the same time, hydrocortisone at a concentration of 14 μM stimulated cell growth by 27%, at 1.4 μM by 37%, at 0.14 μM by 28%, and at 27 μM by 3% [63]. Another report demonstrated that DHEA inhibits the growth of two strains of HeLa and WI-38 cells in culture and that this inhibition can be largely overcome by the addition of a combination of four deoxy- and four ribonucleosides to the culture medium [64]. To date, the effect of androstenes on the viability of PBMCs has not been studied, but their effects on cytokine production and other immune functions have been analyzed, and DHEA has been shown to restore lost or damaged immunity by increasing the responsiveness to inflammatory inducers [65]. Here, we showed that none of the steroids tested influence the viability of PBMCs. Earlier, we showed that progesterone inhibits it by up to 30% [54]. Data on the viability of K562 cells are not available; the only reference is about the influence of DHEA derivatives, but not DHEA itself, on HeLa and K562 cells, where the IC50 of inhibition was 4–95 μM [69].

There are no data about the effects of 5-AED on cell cultures, although it has a strong immunostimulatory effect in vivo [70]. A subcutaneous injection of DHEA protected mice from lethal infections, including both herpes virus type 2 encephalitis and systemic coxsackievirus B 4 (CB 4) infections. 5-AED, a metabolic product of DHEA, was almost a hundred times more effective in regulating the systemic resistance to lethal infection with CB 4 than its precursor, DHEA. In addition to its protective effect, 5-AED, but not DHEA, induced a three- to fourfold increase in cell proliferation in the spleen and the thymus of virus-infected animals; this effect of 5-AED was only seen at doses somewhat exceeding a threshold dose. Neither steroid, however, showed any significant direct antiviral effect in vitro; similarly, virus titers in vivo were not affected by the hormones. These data demonstrate that DHEA and 5-AED actions depend on the expression profiles of their targets in the appropriate cells and tissues. DHEA acts predominantly through ERs and ARs, but AED receptors are still not determined. As DHEA is a precursor of 5-AED, more time and a higher concentration are needed to achieve the same effect as 5-AED. That could be an explanation for why the DHEA-S concentration is the highest among all steroids in humans (up to 10 µM). Under stress conditions, the metabolic pathways leading to 5-AED synthesis could be activated.

The esterification of DHEA and 5-AED can lead to their prolonged circulation, as was demonstrated for pregnane progestin gestobutanoyl [71]. These data demonstrate that the esters of DHEA and 5-AED open up new horizons in protecting people from infectious diseases, cancer, and inflammatory diseases.

The effects of DHEA in vitro may develop within seconds or minutes, which may indicate the occurrence of fast nongenomic effects that are realized through the membrane receptors of sex steroid hormones (SSHs). The effects of steroid hormones are mediated not only via intracellular or membrane-associated receptors (such as the androgen receptor (AR), estrogen receptor alpha (ERα), and estrogen receptor beta (ERβ)) but also via transmembrane receptors (such as zinc transporter protein 9 (ZIP9) and G protein-coupled estrogen receptor 1 (GPER1)). While the classical effects of SSHs via nuclear AR, ERα, and ERβ are relatively well described, the physiological importance of the rapid, non-classical actions of SSHs via membrane-associated (AR, ERα, ERβ) and transmembrane-associated, GPCR steroid receptors (ZIP9, GPER1) is poorly understood [72].

The inhibitory effect of androstenes on the chemiluminescence of mononuclear cells, which indicates their anti-ROS and anti-inflammatory activities, develops very fast, within minutes. It was shown earlier that the cytotoxic action of doxorubicin decreases in the presence of androstenes [52]. The main mechanism of its action is the generation of ROS, and androstenes possibly block/inhibit ROS production. The highest antioxidant effect was shown for DHEA 3-propionate (**3b**), which inhibited LSCL by 73% compared to the control, DHEA, which inhibited it only by 15%. The same compound, DHEA 3-propionate (**3b**), was also effective in inhibiting the growth of HeLa and K-562 cells, indicating its antiproliferative activity. The rational modification of steroid hormones is a well-known and proven strategy for the development of improved treatments of hormone-dependent and other cancers [73]. Promising data were obtained concerning the antiproliferative action of modified androstenes: the fusion of pyridine to position 3,4 of the A-ring can enhance selective antiproliferative activity against PC-3 cells; the A-pyridine D-lactone steroid also exhibited selective submicromolar antiproliferative activity against HT-29 colon cancer cells [74]. The data obtained in the present study show that propionic substituents at positions 3-C and 17-C in steroid molecules hold promise for the creation of immunostimulatory and cytoprotective substances with antioxidant properties.

## 5. Conclusions

It has been shown for the first time that 5-AED and DHEA derivatives at a physiological concentration of 10 μM produce a strong cytotoxic effect on HeLa and K562 cancer cells and do not damage normal Wi-38 and PMBC cells, which suggests that these androstenes have cytoprotective properties toward noncancerous cells. The strongest cytostatic effect during the long-term (5 days) incubation of HeLa and K562 cells was demonstrated by DHEA 3-acetate (**3a**), DHEA 3-propionate (**3b**), and 5-AED 3,17-dipropionate (**4b**). During incubation for 24 h, only DHEA 3-propionate (**3b**) and 5-AED 3,17-dipropionate (**4b**) exhibited a cytostatic effect. The cells studied differ in the expression of NOX subtypes, ER and AR subtypes, and PPAR subtypes. In addition, energy consumption processes in cancerous and noncancerous cells differ dramatically. This could be the reason for the cytotoxic action of androstenes on cancerous cells and for the anabolic, proliferative effect on Wi-38 cells.

All steroid hormone derivatives synthesized also exhibited antioxidant activity by reducing the production of ROS by peripheral blood mononuclear cells from healthy volunteers, as demonstrated in a luminol-stimulated chemiluminescence assay. The suppression of the oxidant activity of non-stimulated mononuclear cells indicates that the androstenes possess anti-ROS and, possibly, anti-inflammatory activities. The highest antioxidant effects were shown for the propionic ester of DHEA. DHEA 3-propionate (**3b**) inhibited luminol-stimulated chemiluminescence by 73% compared to the control, DHEA, which decreased it only by 15% (*p* < 0.05).

Summarizing the data obtained, we suggest that propionic substituents at 3-C and 17-C in steroid molecules are promising in the creation of antitumor agents with antioxidant and cytoprotective properties toward normal cells.

## Data Availability

The data presented in this study are available in this article.

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
