# Peer review of "The Effects of the Steroids 5-Androstenediol and Dehydroepiandrosterone and Their Synthetic Derivatives on the Viability of K562, HeLa, and Wi-38 Cells and the Luminol-Stimulated Chemiluminescence of Peripheral Blood Mononuclear Cells from Healthy Volunteers"

_biomolecules, 2024, doi:10.3390/biom14030373_

Round 1

Reviewer 1 Report (Previous Reviewer 3)

Comments and Suggestions for Authors

I had the opportunity to review the previous version of the manuscript. The authors corrected what I asked for earlier. 

I agree that the MDPI publisher does not require uploading CIF files with deposited structures, but it would have made the review process easier. In any case, I contacted CCDC and the structures are not raising any concerns. 

Comments on the Quality of English Language

Maybe the publisher's correction will be enough to correct the language. 

Reviewer 2 Report (Previous Reviewer 4)

Comments and Suggestions for Authors

No Comments.

This manuscript is a resubmission of an earlier submission. The following is a list of the peer review reports and author responses from that submission.

Round 1

Reviewer 1 Report

Comments and Suggestions for Authors

In this paper, the authors describe a simple and efficient synthesis of esters from starting DHEA, which are known in the literature. They have tested their cytotoxicity against several cancer cell lines and healthy cells and additionally examined the antioxidant activity of the synthesized compounds.

The research is interesting, but I believe that for publication in a higher-ranking journal, additional tests should be done to determine the cytotoxic mechanism of action of the investigated compounds, but even after that, I still think that the work is not within the scope of the topic entitled Design, Synthesis and Biological Evaluation of Novel Small Molecules as Multi-target Enzyme Inhibitors, and I must suggest to the editors to reject this paper.

Additionally, it is unusual for steroid compounds to use their complete IUPAC  names, but more often Semisystematic nomenclature for natural products is used.

Although, the authors claim that there are few publications regarding the effect of androstenes on the viability of various cell cultures, a larger number of works can be found by searching the literature.

And lastly, I was not able to see the supplementary material.

Reviewer 2 Report

Comments and Suggestions for Authors

This work: Effect of steroid hormones 5-AED, DHEA, and their synthetic deriv- atives on the viability of K562, HeLa, Wi-38 cells and the luminol-3 stimulated chemiluminescence of peripheral blood mononuclear cells 4 from healthy volunteers, is a good work

Reviewer 3 Report

Comments and Suggestions for Authors

The submitted manuscript for review describes the synthesis of several acyl derivatives of selected steroid hormones (DHEA and 5-AED) and their biological studies. All of these derivatives are known except 4c (5-AED 3,17-dibutanoate) according to the Reaxys database. The authors provide reference 29 for this compound, in which the physicochemical characterization of 4c is lacking. This is probably why 4c is a new compound according to Reaxys. Sex hormones are highly carcoregenic, so their acyl derivatives should also show cytotoxicity. Therefore, the results are predictable. 

In this review, I will focus primarily on the synthesis and physicochemical characterization, as biological studies are outside the scope of my specialty. 

I will make my remarks according to the order in the text.

1) What purity were the reagents purchased? The term "commercial grade" is incorrect (l. 79). Where were these reagents purchased (company, location)?

2) l. 83. "1% of water" Could KMnO4 be of 99%? This is not possible. 

3) It is not enough to write (a statement repeated several times) l. 88 and 89 that the melting points and specific rotations were "identical" to the values in the literature. It is necessary to give own values and cite values found in the literature with references. 

4) If compounds have abbreviations in the form of numbers, there is no need to write "compound" and add a numerical abbreviation. For example, l. 92 and 93, instead of writing "compound 1," it is better to use only "1." 

5) l. 99. Degree is not a unit of specific rotation. Degree is the unit of torsion of the plane. The unit of specific rotation is quite complex, so in the literature it is given without a unit. 

6) Copies of NMR spectra should be shown in the supporting materials. And already mandatory for the new compound. Elemental analysis (or possibly HRMS) should also be done for the new compound.

7) What do the authors mean when they write "Hydroxy H atoms...". (l. 193)? This is an error. Table S1 is not shown because there are no supporting materials. CIF files and files confirming that the structures were conducted correctly (from CCDC) should be included as review materials. 

8) l. 215 "L-glutamine" Configuration series are written in smaller font. Here it is the same. 

9) l. 242. the uncertainty (here 3.13) can have up to two significant digits. Here it has three. 

10) I don't see the point of citation (l. 250, Ref. 35) since the details of the experiment are given. It is self-citing. It should be removed because it is not needed.

11) l. 275. Scheme 1 is not mentioned in the text, it appears at the beginning of the chapter. There should be text first and necessarily a reference in the text to this object. 

12) The text regarding synthesis and single-crystal X-ray diffraction is perfunctory. 

13) Table 1. Uncertainty can have a maximum of two significant digits. Here it often has as many as three significant digits. An explanation of uncertainty rounding can be found here, for example: https://www.utsc.utoronto.ca/~quick/PHYB10H3F/Lectures/PHYB1009.pdf or here: https://www.physics.upenn.edu/sites/default/files/Managing%20Errors%20and%20Uncertainty.pdf

14) l. 481 The statement that the compounds tested are new is incorrect. Only one is new. "...new 5-AED and DHEA derivatives...".

Comments on the Quality of English Language

Language correction of the manuscript is needed. Inappropriate choice of words, for example, l. 86 "correspondently" instead of " respectively." 

Reviewer 4 Report

Comments and Suggestions for Authors

This paper reports the synthesis of seven dehydroepiandrosterone derivatives from dehydroepiandrosterone, which exhibited micromolar toxicity in Hela and K526 cancer cells while stimulating growth-promoting activity in non-cancerous cells. Moreover, the derivatives could show superior antioxidant effects by reducing the production of reactive oxygen species in peripheral blood mononuclear cells. The article is innovative. Therefore, I support its publication in Biomolecules with minor modifications. The changes are as follows

1. Please provide a picture of the C/H spectrum of the NMR

2. Whether the synthesized compound is a new compound, and whether other compounds (including intermediates) can provide other proof except compounds whose structure is confirmed by single crystal diffraction; Especially 3c, the hydrogen spectrum is obviously wrong

3. The Introduction is too messy, please re-summarize each paragraph and make a summary.

4. Can you explain the stimulating effect of 347 corticosteroids on Wi-38

5. The steroid in 3.2.1 has almost no toxicity to normal cells, while it is more toxic to hela cells. May I ask what causes it? Is it possible that it could have been caused by something other than the recipient.

6. In row 398, metabolites of steroids play an important role, can this metabolite be produced in all cells? 5-AED as one of the metabolites of steroids, the effect is a hundred times that of steroids, is it possible that the concentration of steroids is too high

Comments on the Quality of English Language

Describe the experimental results using concise and precise language, avoiding unnecessary words.